# Push-pull Feedback Implements Hierarchical Information Retrieval Efficiently

**Xiao Liu**[1]        **Xiaolong Zou**[1]        **Zilong Ji**[2]        **Gengshuo Tian**[3]

**Yuanyuan Mi**[4]        **Tiejun Huang**[1]        **K. Y. Michael Wong**[5]

**Si Wu**[1]

[1]School of Electronics Engineering & Computer Science, IDG/McGovern Institute for Brain Research,
Peking-Tsinghua Center for Life Sciences, Academy for Advanced Interdisciplinary Studies,
Peking University, Beijing, China.
[2]State Key Laboratory of Cognitive Neuroscience and Learning, Beijing Normal University, China.
[3]Department of Mathematics, Beijing Normal University, China.
[4]Center for Neurointelligence, Chongqing University, China.
[5]Department of Physics, Hong Kong University of Science and Technology, China.
{xiaoliu23,xiaolz,tjhuang,siwu}@pku.edu.cn,
jizilong@mail.bnu.edu.cn, gengshuo_tian@163.com,
miyuanyuan0102@cqu.edu.cn, phkywong@ust.hk

## Abstract

Experimental data has revealed that in addition to feedforward connections, there exist abundant feedback connections in a neural pathway. Although the importance of feedback in neural information processing has been widely recognized in the field, the detailed mechanism of how it works remains largely unknown. Here, we investigate the role of feedback in hierarchical information retrieval. Specifically, we consider a hierarchical network storing the hierarchical categorical information of objects, and information retrieval goes from rough to fine, aided by dynamical push-pull feedback from higher to lower layers. We elucidate that the push (positive) and pull (negative) feedbacks suppress the interferences due to neural correlations between different and the same categories, respectively, and their joint effect improves retrieval performance significantly. Our model agrees with the push-pull phenomenon observed in neural data and sheds light on our understanding of the role of feedback in neural information processing.

## 1 Introduction

Deep neural networks (DNNs), which mimic hierarchical information processing in the ventral visual pathway, have achieved great success in object recognition [15]. The structure of DNNs mainly contains feedforward connections from lower to higher layers. The experimental data, however, has revealed that there also exist abundant feedback connections from higher to lower layers, whose number is even larger than that of feedforward ones [23]. It has been widely suggested that these feedback connections play an important role in visual information processing. For instance, the theory of analysis-by-synthesis proposes that the feedback connections, in coordination with the feedforward ones, enable the neural system to recognize an object in an interactive manner [16], that is, the feedforward pathway extracts the object information from external inputs, while the feedback pathway generates hypotheses about the object; and the interaction between the two pathways

accomplishes the recognition task. Based on a similar idea, the theory of predictive coding proposes that the feedback from the higher cortex predicts the output of the lower cortex [22]. Although the importance of feedback has been widely recognized in the field, computational models elucidating how it works exactly remain poorly developed. Interestingly, the experiment data has unveiled a salient characteristic of feedback in the visual system [8, 6]. Fig.1A displays the neural population activities in V1 when a monkey was performing a contour integration task [8]. In response to the visual stimulus, the neural activity in V1 increased at the early phase, displaying the push characteristic; and decreased at the late phase, displaying the pull characteristic. Multi-unit recording revealed that in the pull phase, there was strong negative feedback from the higher cortex (V4) [6]; while in the push phase, although the contributions of the feedforward input and feedback were mixed, causality analysis confirmed that there indeed existed a feedback component [7].

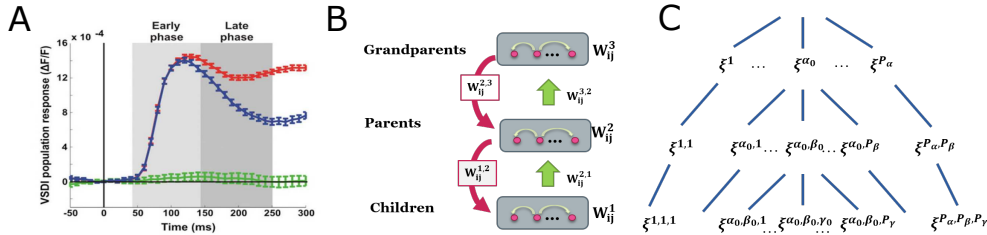

Figure 1: A. The push-pull phenomenon of neural population activity. Data was recorded at V1 of an awake monkey. The visual stimulus was a contour embedded in noises, which was onset at $t = 0$. The blue curve shows the change of neural response over time, which increased at the early phase and decreased at the late phase, when the monkey recognized the contour. The red curve represents the condition when the monkey did not recognize the contour, and the green curve the condition of no visual stimulus. Adopted from [8]. B. A three-layer network storing hierarchical categories of objects, denoted, from bottom to top, as child, parent, and grandparent patterns, respectively. Between layers, neurons are connected by both feedforward and feedback connections. C. The branching tree displaying the categorical relationships between hierarchical memory patterns.

The categorization of objects is based on their similarity/dissimilarity either at the image level or in the semantic sense, and it is organized hierarchically, in the sense that objects belonging to the same category are more similar than those belonging to different ones. The experimental data has revealed that the brain encodes these similarities using overlapping neural representations, with the greater the similarity, the larger the correlation between neural responses [26]. However, it is well known that neural networks have difficulties of storing and retrieving correlated memory patterns; a small amount of correlation in the Hopfield model will deteriorate memory retrieval dramatically [9]. Notably, this instability is intrinsic to a neural network, as it utilizes synapses to carry out both information encoding and retrieving: the synaptic strengths are affected by the correlations of stored patterns, which in turn interfere with the retrieval of a stored pattern. Thus, a dilemma is raised to neural coding: on one hand, to encode the categorical relationships between objects, a neural system needs to exploit correlated neural representations; on the other hand, to retrieve information reliably, these patterns correlations are harmful. How to achieve reliable hierarchical information retrieval in a neural network remains unresolved in the field [2, 13].

In the present study, motivated by the push-pull phenomenon in neural data, we investigate the role of feedback in hierarchical information retrieval. Specifically, we consider that a neural system employs a rough-to-fine retrieval procedure, in which higher categorical representations of objects are retrieved first, since they are less correlated than lower categorical ones and hence have better retrieval accuracy; subsequently, through feedback, the retrieved higher categorical information facilitates the retrieval of lower categorical representations. We elucidate that the optimal feedback should be dynamical, varying from positive (push) to negative (pull) over time, and they suppress the interferences due to pattern correlations from different and the same categories, respectively. Using synthetic and real data, we demonstrate that the push-pull feedback implements hierarchical information retrieval efficiently.

## 2 A model for Hierarchical Information Representation

To elucidate the role of feedback clearly, we consider a simple model for hierarchical information representation. The model is kept simple to illustrate insights derived from the role played by different sources of interferences during different stages of dynamical retrieval. Specifically, the model consists of three layers which store three-level hierarchical memory patterns. For convenience, we call the three layers, from bottom to top, child, parent, and grandparent layers, respectively, to reflect their ascending category relationships (Fig.1B). Neurons in the same layer are connected recurrently with each other to function as an associative memory. Between layers, neurons communicate via feedforward and feedback connections. Denote the state of neuron $i$ in layer $l$ at time $t$ as $x_i^l(t)$ for $i = 1, \ldots, N$, which takes value of $\pm 1$, the symmetric recurrent connections from neuron $j$ to $i$ in layer $l$ as $W_{ij}^l$, the feedforward connections from neuron $j$ of layer $l$ to neuron $i$ in layer $l + 1$ as $W_{ij}^{l+1,l}$ and the feedback connections from neuron $j$ of layer $l + 1$ to neuron $i$ of layer $l$ as $W_{ij}^{l,l+1}$. The neuronal dynamics follows the Hopfield model [10], which is written as

$$x_i^l(t + 1) = \text{sign} \left[ h_i^l(t) \right], \tag{1}$$

where $\text{sign}(h) = 1$ for $h > 0$ and $\text{sign}(h) = -1$ otherwise. $h_i^l(t)$ is the total input received by the neuron, which is given by (only the result for layer 1 is shown, and the results for other layers are similar),

$$h_i^1(t) = \sum_j W_{ij}^1 x_j^1(t) + \sum_j W_{ij}^{1,2} x_j^2(t). \tag{2}$$

We generate synthetic patterns to study information retrieval, which are denoted as: $\{\xi^\alpha\}$ for $\alpha = 1, \ldots, P_\alpha$ represents the grandparent patterns, $\{\xi^{\alpha,\beta}\}$ for $\beta = 1, \ldots, P_\beta$ the parent patterns of grandparent $\alpha$, and $\{\xi^{\alpha,\beta,\gamma}\}$ for $\gamma = 1, \ldots, P_\gamma$ the child patterns of parent $(\alpha, \beta)$, where $P_\alpha, P_\beta$, and $P_\gamma$ are denoted as the number of grandparent patterns, the number of parent patterns belonging to the same grandparent, and the number of child patterns belonging to the same parent, respectively. These hierarchical memory patterns are constructed as follows [1] (Fig.1C).

First, grandparent patterns are statistically independent of each other. The value of each element in a grandparent pattern is drawn from the distribution

$$P(\xi_i^\alpha) = \frac{1}{2}\delta(\xi_i^\alpha + 1) + \frac{1}{2}\delta(\xi_i^\alpha - 1), \tag{3}$$

where $\delta(x) = 1$ for $x = 0$ and $\delta(x) = 0$ otherwise. Each element of a grandparent has equal probabilities of taking a value of 1 or $-1$.

Secondly, for each grandparent pattern, its descending parent patterns are drawn from the distribution

$$P(\xi_i^{\alpha,\beta}) = (\frac{1 + b_2}{2})\delta(\xi_i^{\alpha,\beta} - \xi_i^\alpha) + (\frac{1 - b_2}{2})\delta(\xi_i^{\alpha,\beta} + \xi_i^\alpha), \tag{4}$$

where $0 < b_2 < 1$ implies that each element of a parent pattern has a probability of $(1 + b_2)/2 > 0.5$ to have the same value as the corresponding element of the grandparent. This establishes the relationship between a grandparent and the parent patterns.

Thirdly, for each parent pattern, its descending child patterns are drawn from the distribution

$$P(\xi_i^{\alpha,\beta,\gamma}) = (\frac{1 + b_1}{2})\delta(\xi_i^{\alpha,\beta,\gamma} - \xi_i^{\alpha,\beta}) + (\frac{1 - b_1}{2})\delta(\xi_i^{\alpha,\beta,\gamma} + \xi_i^{\alpha,\beta}), \tag{5}$$

where $0 < b_1 < 1$, which specifies the relationship between a parent and its child patterns.

The above stochastic pattern generation process specifies the categorical relationships among memory patterns, in the sense that patterns in the same group have stronger correlation than those belonging to different groups. For example, the correlation between two child patterns belonging to the same parent (siblings) is given by $\sum_i \xi^{\alpha,\beta,\gamma}\xi^{\alpha,\beta,\gamma'}/N = b_1^2$, referred to as the intra-class correlation; the correlation between two child patterns belonging to different parents but sharing the same grandparent (cousins) is given by $\sum_i \xi^{\alpha,\beta,\gamma}\xi^{\alpha,\beta',\gamma'}/N = b_1^2 b_2^2$, referred to as the inter-class correlation; and the correlation between two child patterns belonging to different grandparents is given by $\sum_i \xi^{\alpha,\beta,\gamma}\xi^{\alpha',\beta',\gamma'}/N = 0$. These correlation values satisfy the hierarchical relationship, i.e.,

$b_1^2 > b_1^2 b_2^2 > 0$. Other correlation relationships can be obtained similarly (see Sec.1 in Supplementary Information (SI)).

Each layer of the network behaves as an associative memory. Using the standard Hebbian learning rule, the recurrent connections between neurons in the same layer are constructed to be $W_{ij}^1 = \sum_{\alpha,\beta,\gamma} \xi_i^{\alpha,\beta,\gamma} \xi_j^{\alpha,\beta,\gamma}/N$, $W_{ij}^2 = \sum_{\alpha,\beta} \xi_i^{\alpha,\beta} \xi_j^{\alpha,\beta}/N$, and $W_{ij}^3 = \sum_{\alpha} \xi_i^{\alpha} \xi_j^{\alpha}/N$. The feedforward connections from lower to higher layers are set to be $W_{ij}^{2,1} = \sum_{\alpha,\beta,\gamma} \xi_i^{\alpha,\beta} \xi_j^{\alpha,\beta,\gamma}/N$, and $W_{ij}^{3,2} = \sum_{\alpha,\beta} \xi_i^{\alpha} \xi_j^{\alpha,\beta}/N$. It is easy to understand the effect of feedforward connections. For example, if layer 1 is at the state of the memory pattern $\xi^{\alpha_0,\beta_0,\gamma_0}$, then the feedforward input to layer 2 is given by $\sum_j W_{ij}^{2,1} \xi_j^{\alpha,\beta,\gamma} \approx \xi_i^{\alpha_0,\beta_0}$, which contributes to improving the retrieval of the parent pattern $\xi^{\alpha_0,\beta_0}$ at layer 2. The form of feedback connections is the focus of this study and will be introduced later.

To quantify the retrieval performance, we define a macroscopic variable $m(t)$, measuring the overlap between the neural state $\mathbf{x}(t)$ and a memory pattern, which is calculated to be [9] (again, for simplicity, only the result for layer 1 is shown),

$$m^{\alpha,\beta,\gamma}(t) = \frac{1}{N} \sum_{i=1}^{N} \xi_i^{\alpha,\beta,\gamma} x_i^1(t). \tag{6}$$

where $-1 < m^{\alpha,\beta,\gamma}(t) < 1$ represents the retrieval accuracy of the memory pattern $\xi^{\alpha,\beta,\gamma}$, and the larger the value of $m$, the higher the retrieval accuracy.

## 3 Information retrieval without feedback

To elucidate the role of feedback, it is valuable to first check information retrieval without feedback, and without loss of generality, we focus on layer 1. Following the standard stability analysis [9], we consider that the initial state of layer 1 is a memory pattern, $\mathbf{x}^1(0) = \xi^{\alpha_0,\beta_0,\gamma_0}$, and investigate what are the key factors determining the retrieval performance. After one step of iteration, we get the retrieval accuracy,

$$
\begin{aligned}
m^{\alpha_0,\beta_0,\gamma_0}(1) &= \frac{1}{N} \sum_{i=1}^{N} \xi_i^{\alpha_0,\beta_0,\gamma_0} x_i^1(1) = \frac{1}{N} \sum_{i=1}^{N} \xi_i^{\alpha_0,\beta_0,\gamma_0} \mathrm{sign}\left[ h_i^1(0) \right], \\
&= \frac{1}{N} \sum_{i=1}^{N} \mathrm{sign}\left[ \xi_i^{\alpha_0,\beta_0,\gamma_0} h_i^1(0) \right].
\end{aligned} \tag{7}
$$

We see that the retrieval of a memory pattern is determined by the alignment between the neural input and the memory pattern, which is further written as (see Sec.2 in SI),

$$\xi^{\alpha_0,\beta_0,\gamma_0} h_i^1(0) = \xi^{\alpha_0,\beta_0,\gamma_0} \frac{1}{N} \sum_j W_{ij}^1 x_j^1(0) = 1 + C_i + \widetilde{C}_i. \tag{8}$$

Here, the input received by the neuron is decomposed into the signal and noise parts, and the latter is further divided into two components, $C_i$ and $\widetilde{C}_i$, which represent, respectively, the interferences to memory retrieval due to: 1) the correlation of the pattern to be retrieved with siblings from the same parent, called the **intra-class interference**; 2) the correlation of the pattern to be retrieved with cousins from the same grandparent but different parents, called the **inter-class interference**. It can be checked that in the limits of large $N$, $P_\gamma$ and $P_\beta$, the intra- and inter- class interferences, $C_i$ and $\widetilde{C}_i$, satisfy the distributions, $P(C_i) = \mathcal{N}(E_C, V_C)(1 + b_1)/2 + \mathcal{N}(-E_C, V_C)(1 - b_1)/2$, $P(\widetilde{C}_i) = \mathcal{N}(E_{\widetilde{C}}, V_{\widetilde{C}})(1 + b_1 b_2)/2 + \mathcal{N}(-E_{\widetilde{C}}, V_{\widetilde{C}})(1 - b_1 b_2)/2$, where $\mathcal{N}(E,V)$ represents a normal distribution with mean $E$ and variance $V$, and $E_C = b_1^3(P_\gamma - 1)$, $E_{\widetilde{C}} = b_1^3 b_2^3 P_\gamma (P_\beta - 1)$, $V_C = b_1^4(P_\gamma - 1)(1 - b_1^2)$, $V_{\widetilde{C}} = b_1^4 b_2^4 P_\gamma (P_\beta - 1)(1 - b_1^2)(1 - b_2^2)$ (see Sec.2 in SI).

The breadth of the above noise distributions, as a consequence of pattern correlations, implies that even starting from a noiseless state, the network dynamics still incur retrieving instability [9], and the error occurs when noises are large (i.e., $C_i + \widetilde{C}_i < -1$).

# 4 Hierarchical Information Retrieval with the push-pull feedback

According to the above theoretical analysis, to improve memory retrieval, the key is to suppress the inter- and intra- class noises due to pattern correlations. Note that, in practice, the correlations between higher categorical patterns tend to be smaller than that between lower categorical patterns. For example, the similarity between cats and dogs is usually smaller than that between two sub-types of cats. In our model, this corresponds to the condition of $b_1 > b_2$. For an associative memory, this implies that given the same amount of input information (e.g., an ambiguous image of a Siamese cat), the parent pattern (e.g., a cat) can be better retrieved than the child pattern (e.g., a Siamese cat). Thus, we consider a rough-to-fine retrieval procedure, in which the parent pattern in layer 2 is first retrieved, whose result is subsequently fed back to layer 1 to improve the retrieval of the child pattern.

Below, for the convenience of analysis, we assume that the parent pattern is first perfectly retrieved ($m = 1$) and explore the appropriate form of feedback which can efficiently utilize the parent information to enhance the retrieval of the child pattern. Later, we carry out simulations demonstrating that the model works in general cases when the parent pattern is not perfectly retrieved.

## 4.1 The form and the role of push feedback

We first show that a push (positive) feedback of a proper form can suppress the inter-class interference in memory retrieval effectively. Without loss of generality, we consider that for a given input, the corresponding child pattern to be retrieved in layer 1 is $\xi^{\alpha_0,\beta_0,\gamma_0}$ and that the corresponding parent pattern in layer 2 is $\xi^{\alpha_0,\beta_0}$. Consider the push feedback of the below form,

$$W_{ij}^{1,2} = \frac{1}{NP_\gamma} \sum_{\alpha,\beta,\gamma} \xi_i^{\alpha,\beta,\gamma} \xi_j^{\alpha,\beta}, \tag{9}$$

which follows the standard Hebb rule between the parent and their child patterns, and its contribution is intuitively understandable. Given that the parent pattern $\xi^{\alpha_0,\beta_0}$ in layer 2 is retrieved, its push feedback to neuron $i$ in layer 1 is calculated to be $\sum_j W_{ij}^{1,2} \xi_j^{\alpha_0,\beta_0} \approx \sum_\gamma \xi_i^{\alpha_0,\beta_0,\gamma}/P_\gamma$. Obviously, this positive current increases the activities of all child patterns belonging to the parent, i.e., those $\xi^{\alpha_0,\beta_0,\gamma}$ for any $\gamma$, and it has little influence on other child patterns from different parents, i.e., those $\xi^{\alpha_0,\beta,\gamma}$ for $\beta \neq \beta_0$. Due to the competition between memory patterns in the network dynamics, this effectively suppresses the inter-class interference in memory retrieval (Fig.2A ).

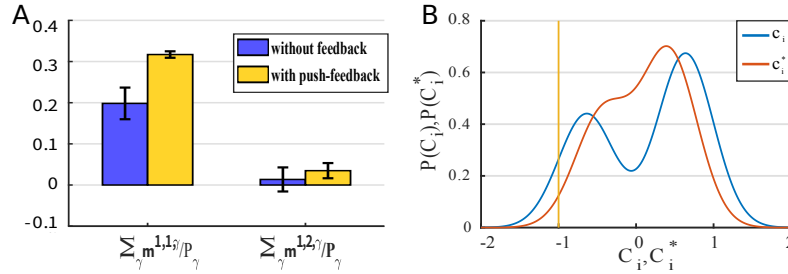

Figure 2: A. Illustrating the effect of push-feedback. The parent pattern is $\xi^{1,1}$, whose feedback contribution to sibling patterns is measured by $\sum_\gamma m^{1,1,\gamma}/P_\gamma$, and to cousin patterns is measured by $\sum_\gamma m^{1,2,\gamma}/P_\gamma$. The results averaged over 100 trials are shown. B. Illustrating the effect of pull-feedback. The distributions of the intra-class noise $C_i$ without feedback and the noise $C_i^*$ with the pull-feedback are presented (see Eqs.(11,14) in SI). Retrieval errors occur when $C_i < -1$ or $C_i^* < -1$ (indicated by the yellow line). The parameters are $N = 2000, P_\alpha = 2, P_\beta = 10, P_\gamma = 70, b_1 = 0.2$, and $b_2 = 0.15$.

## 4.2 The form and the role of pull feedback

We further show that a pull (negative) feedback of an appropriate form can suppress the intra-class interference in memory retrieval. Consider the pull feedback of the below form,

$$W_{ij}^{1,2} = -b_1 \delta_{ij}. \tag{10}$$

Given the parent pattern $\xi^{\alpha_0,\beta_0}$ in layer 2 is retrieved, its negative feedback to neuron $i$ in layer 1 is calculated to be $\sum_j W_{ij}^{1,2}\xi_j^{\alpha_0,\beta_0} = -b_1\xi_i^{\alpha_0,\beta_0}$. In the large $P_\gamma$ limit, the parent pattern is approximated to be the mean of its child patterns (Sec.1 in SI), thus, in effect the pull feedback is to subtract a portion of the mean value from sibling patterns. The retrieval accuracy of the target child pattern after applying the pull feedback is calculated to be (Sec.3 in SI),

$$m^{\alpha_0,\beta_0,\gamma_0} = \frac{1}{N}\sum_{i=1}^{N}\text{sign}\left[1 + C_i^* + \widetilde{C}_i\right], \tag{11}$$

where $C_i^* \equiv C_i - b_1\xi_i^{\alpha_0,\beta_0,\gamma_0}\xi_i^{\alpha_0,\beta_0}$ is the new noise term after applying the pull-feedback. As shown in Fig.2B, with the pull feedback, the negative tail of the noise distribution (where retrieval errors occur) is considerably reduced.

### 4.3 The joint effect of the push-pull feedback

Summarizing the above results, we come to the conclusion that to achieve good information retrieval, the neural feedback needs to be dynamical, exerting the push and pull components at different stages, so that they can suppress the inter- and intra- class interferences, respectively.

To better demonstrate the joint effect of the push-pull feedback, we consider a continuous version of the Hopfield model, so that the network state changes smoothly and the joint effects of push- and pull- feedbacks are integrated over time (the discrete Hopfield model still works, but the overall effect is less significant). The network dynamics are given by [11]

$$\tau\frac{dh_i^n}{dt} = -h_i^n + \sum_j W_{ij}^n x_j^n + \sum_k W_{ik}^{n,m}(t)x_k^m + I_i^{ext,n}, \tag{12}$$

$$x_i^n = f(h_i^n), \quad m,n = 1,2, \tag{13}$$

where $h_i^n$ and $x_i^n$ denote the synaptic input and the firing rate of neuron $i$ in layer $n$, respectively, and their relationship is given by a sigmoid function, $f(x) = \arctan(8\pi x)/\pi + 1/2$, therefore $0 < x_i^n < 1$. To match the strength of firing rate $x_i^n$, we also align all the hierarchical patterns $\xi_i$ into $0,1$. The parameter $\tau$ is the time constant. The recurrent and feedforward connections follow the standard Hebb rule as described above. The feedback connections are slightly modified from Eqs.(9-10) to accommodate positive values of neural activities in the continuous model. They are given by: the push-feedback $W_{ik}^{1,2} = a_+ P_\gamma \sum_{\alpha,\beta,\gamma}(\xi_i^{\alpha,\beta,\gamma} - \langle\xi\rangle)(\xi_k^{\alpha,\beta,\gamma} - \langle\xi\rangle)/N$, with $a_+$ being a positive number, and the pull-feedback $W_{ik}^{1,2} = -a_- b_1\delta_{ik}$, with $a_-$ being a positive number. $I_i^{ext}$ is the external input conveying the object information. The push and pull feedbacks are applied sequentially, with each of them lasting in the time order of $\tau$ ($\tau \sim 10 - 20$ms), as suggested by the data [6]. For details of the model, see Sec.4.1 in SI.

Fig.3 displays a typical example of the memory retrieval process in the network, demonstrating that: 1) the neural population activity at layer 1 exhibits the push-pull phenomenon, agreeing qualitatively with the experimental data (Fig.3A compared to Fig.1A); 2) the retrieval accuracy of layer 1 with the push-pull feedback is improved significantly compared to the case of no feedback (Fig.3B). Interestingly, we note that when the push feedback is applied, the retrieval accuracy of the target child pattern is decreased a little bit. This is due to that the push feedback only aims at reducing the inter-class interference without differentiating sibling patterns.

We evaluate the performances of the model by varying the amplitude of pattern correlations and confirm that the push-pull feedback always improves the network performance statistically (Sec.4.2 in SI).

## 5 Applying to Real Images

We test our model in the processing of real images. As shown in the top of Fig.4A, the dataset we use consists of $P_\beta = 2$ types of animals, cats and dogs, corresponding to parents in our model. For each type of animal, it is further divided into $P_\gamma = 9$ sub-types, corresponding to children. A total of $1800$ images, with $K = 100$ for each sub-type of animals, are chosen from ImageNet. It has been shown that the neural representations generated by a DNN (after being trained with ImageNet) capture the

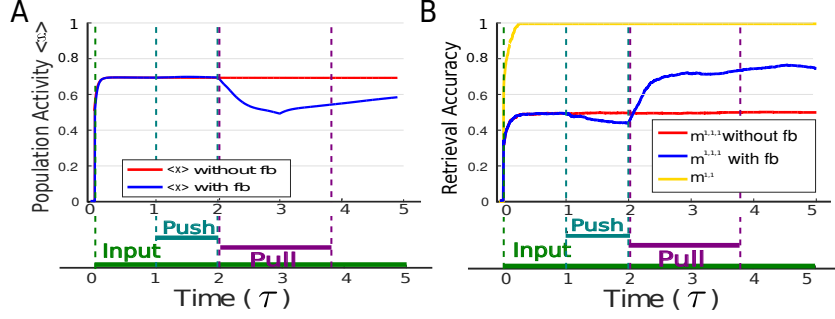

Figure 3: A. The neural population activity $\langle x \rangle$ in layer 1 as a function of time in a typical trial (blue curve), which exhibits the push-pull phenomenon as observed in the experiment [8]. The red curve is the case without feedback. $\langle x \rangle$ is obtained by averaging over all neurons in layer 1. B. The retrieval accuracies of the child (red curve) and the parent patterns (yellow curve) as functions of time in the same trial as in A. The blue curve is the case without feedback. The lower panel in both A and B displays the time course of applying an external input ($t \in (0, 4\tau)$), the push-feedback ($t \in (\tau, 2\tau)$), and the pull-feedback ($t \in (2\tau, 3\tau)$). The child pattern conveyed by the external input is $\xi^{1,1,1}$, and the corresponding parent pattern is $\xi^{1,1}$. The parameters used are: $N = 2000$, $P_\gamma = 25$, $P_\beta = 4$, $P_\alpha = 2$, $b_1 = 0.2$, $b_2 = 0.1$, $\tau = 5$, $a_{ext} = 1$, $a_r^1 = 1$, $a_r^2 = 2$, $a_{ext}^2 = 0.1$, $a_+ = 1$, $a_- = 10$, $\lambda_1$=0.1, $\lambda_2$=0.1.

categorical relationships between objects, in the sense that the overlap between neural representations reflect the closeness of objects in category, rather than their similarity in pixels [26]. This indicates that the memory patterns are hierarchically organized. We therefore pre-process images by filtering them through VGG, a type of DNN [24], and use the neural representations generated by VGG (i.e., the neural activities before the read-out layer) to construct the memory patterns. The details of pre-processing are described in Sec.5 in SI. The lower panel of Fig.4A shows the correlations between the memory patterns generated by VGG, which exhibits a hierarchical structure, i.e., siblings from the same parent have stronger correlations than cousins from different parents, similar to the correlation structure in our model.

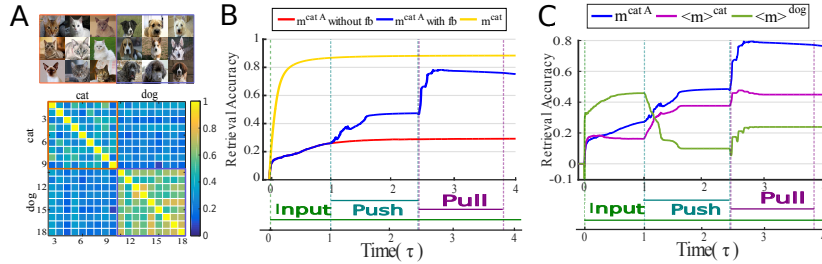

Figure 4: The model performances with real images. A. The dataset. Top panel: example images, one for each sub-type of cat or dog. Lower panel: the correlations between child patterns after pre-processing by VGG. Cat: $1 - 9$; Dog: $10 - 18$. B. Retrieval accuracies of the child (cat A, blue curve) and parent (cat, yellow curve) patterns as functions of time in a typical trial. The red curve is the case without feedback. C. Different effects of the push and pull feedbacks in the example trial as in B. The blue, purple, and green curves represent, respectively, the retrieval accuracies of the target child (cat A, a Siamese cat), the siblings (other sub-types of cats), and other child patterns (all sub-types of dogs) in layer 1. In B-C, the image presented to the network is cat Siamese, and the lower panel displays the time course of applying the external input $(0, 4\tau)$, the push feedback $(\tau, 2.4\tau)$, and the pull feedback $(2.4\tau, 3.8\tau)$. The parameters: $N = 4096$, $a_r^1 = 1$, $a_r^2 = 2$, $a_{ext}^1 = 6$, $a_{ext}^2 = 1$, $a_+ = 2$, $a_- = 1.5$, $a^{21} = 1$. Other parameters are the same as in Fig.3.

We present each image to the network and measure its retrieval accuracy by calculating $m^{\beta,\gamma}$, i.e., the overlap between the network response and the memory pattern corresponding to the image. Fig.4B shows a typical example of the retrieval process. We see that the retrieval accuracy of layer 1 keeps

increasing when the push and pull feedbacks are applied sequentially, and the result is significantly improved compared to the case without feedback. Over $1800$ images, the averaged improvement is $71.04\%$ (measured at the moment when the pull feedback stops).

To illustrate the individual effects of the push and pull feedbacks, we also calculate the retrieval accuracies of sibling and cousin patterns. As shown in Fig.4C, we see that: 1) at the early phase of push feedback, both the retrieval accuracies of the target child pattern and its siblings increase, whereas the retrieval accuracy of cousins drops, indicating that the push feedback has the effect of suppressing the inter-class interference; 2) at the later phase of pull feedback, the retrieval accuracy of the target child pattern experiences another significant increase much larger than that for other child patterns, indicating that the pull feedback has the effect of suppressing the intra-class interference.

## 6  Conclusion and Discussion

The present study investigates the role of feedback in hierarchical information retrieval. Hierarchical associative memory models have been studied previously [21, 1, 25, 20], but these works considered only a single layer network without feedback. To our knowledge, our paper is the first one studying the contributions of feedback. In machine learning, there were studies which utilize the semantics-based higher category knowledge of objects as side information to enhance image recognition [14, 19, 3], but they are very different from our network model in the use of dynamical feedback between layers to enhance information retrieval.

Feedback connections have been widely observed in neural signalling pathways, but their exact computational functions remain largely unclear. Here, in the task of information retrieval, our study reals that the neural feedback, which varies from positive (push) to negative (pull) over time, contributes to the suppression of the inter- and intra- class noises in information retrieval. This push-pull characteristic agrees with the push-pull phenomenon of neural activities observed in the experiments [8, 6]. Notably, the neural systems have resources to realize such a dynamical feedback, and they are likely implemented via different signal pathways. For instance, the push feedback may be realized via direct excitatory synapses from higher to lower layers, and the stopping of push feedback can be controlled by short-term synaptic depression; on the other hand, the pull feedback may go through a separate path mediated by inhibitory interneurons, which is naturally delayed compared to the direct excitatory path [12].

Through studying feedback, the present study also addresses a dilemma in neural coding, which concerns the conflicting roles of neural correlation: on one hand, pattern correlations are essential to encode the categorical relationships between objects; on the other hand, they inevitably incur interference to memory retrieval. To diminish the correlation interference, we propose that neural systems employ a rough-to-fine information retrieval procedure. Upon receiving the external information, the higher categorical pattern is first retrieved, whose result is subsequently utilized to enhance the retrieval of the lower categorical pattern via dynamical push-pull feedback. In such a way, the highly correlated neural representations for objects are reliably retrieved. The idea of rough-to-fine information retrieval is in agreement with the concept of "global first" in cognitive science, which states that the brain extracts first the global (e.g., topological), rather than the local (e.g., Euclidean), geometrical features of objects [4]. This phenomenon has been confirmed by a large volume of psychophysical experiments [5]. Here, our study unveils a computational advantage of "global first" not realized previously, that is, extracting global features first, aided by the push-pull feedback, serves as an efficient strategy to overcome the interference due to neural correlations. It has been suggested by experimental findings that the dorsal pathway [17], and/or the subcortical pathway from retina to superior colliculus [18], carry out the rapid computation of extracting global features of objects; while, along the ventral pathway, the push-pull feedback assists the feedforward input to extract the fine structures of objects in a relatively slow manner. In our future work, we will extend the present study to explore the role of feedback in biologically more detailed models.

**Acknowledgments**

This work was supported by BMSTC (Beijing municipal science and technology commission) under grant No: Z171100000117007 (D.H. Wang & Y.Y. Mi), the National Natural Science Foundation of China (N0: 31771146, 11734004, Y.Y. Mi),the National Natural Science Foundation of China (N0: 61425025, T.J. Huang) Beijing Nova Program (N0: Z181100006218118, Y. Y. Mi), Guangdong Province with Grant (No. 2018B030338001, Si Wu & Y.Y. Mi) and grants from the Research Grants Council of Hong Kong (grant numbers 16322616, 16306817 and 16302419, K. Y. Michael Wong). This work received support from Huawei Technology Co., Ltd..

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
