[Supplementary Material · SI_Push_pull_Feedback_Implements_Hierarchical_Associative_Memory-Final.pdf]

# Push-pull Feedback Implements Hierarchical Information Processing Efficiently: Supplementary Information

## 1 The correlation structure of hierarchical memory patterns

For the hierarchical memory patterns defined in the main text, the following relationships hold:

- The correlation between grandparent patterns, $1/N \sum_i \xi_i^\alpha \xi_i^{\alpha'} = 0$, for $\alpha \neq \alpha'$;

- The correlation between grandparent and its descending parent patterns, $1/N \sum_i \xi_i^\alpha \xi_i^{\alpha,\beta} = b_2$;

- The correlation between parent patterns from the same grandparent, $1/N \sum_i \xi_i^{\alpha,\beta} \xi_i^{\alpha,\beta'} = b_2^2$, for $\beta \neq \beta'$;

- The correlation between parent patterns from different grandparents, $1/N \sum_i \xi_i^{\alpha,\beta} \xi_i^{\alpha',\beta'} = 0$, for $\alpha \neq \alpha'$;

- The correlation between parent pattern and its descending child patterns, $1/N \sum_i \xi_i^{\alpha,\beta} \xi_i^{\alpha,\beta,\gamma} = b_1$;

- The correlation between child patterns from the same parent pattern, $1/N \sum_i \xi_i^{\alpha,\beta,\gamma} \xi_i^{\alpha\beta\gamma'} = b_1^2$, for $\gamma \neq \gamma'$;

- The correlation between child patterns from different parents, $1/N \sum_i \xi_i^{\alpha,\beta,\gamma} \xi_i^{\alpha,\beta',\gamma'} = b_2^2 b_1^2$, for $\beta \neq \beta'$.

Moreover, it can be checked that when the number of parent patterns $P_\beta$ is sufficiently large, the average of parent patterns approaches to their ascending grandparent pattern, i.e., $1/P_\beta \sum_{\beta=1}^{P_\beta} \xi_i^{\alpha,\beta} \approx b_2 \xi_i^\alpha$.

Similarly, when the number of child patterns $P_\gamma$ is sufficiently large, the average of child patterns approaches to their ascending parent pattern, i.e., $1/P_\gamma \sum_{\gamma=1}^{P_\gamma} \xi_i^{\alpha,\beta,\gamma} \approx b_1 \xi_i^{\alpha,\beta}$.

# 2 Memory retrieval without feedback

## 2.1 The neuronal input at layer 1 without feedback

According to Eq.(2) in the main text, without feedback, we have

$$
\begin{aligned}
h_i^1(0) &= \frac{1}{N} \sum_j W_{ij}^1 x_j^1(0), \\
&= \frac{1}{N} \sum_{\alpha,\beta,\gamma} \xi_i^{\alpha,\beta,\gamma} \sum_j \xi_j^{\alpha,\beta,\gamma} \xi_j^{\alpha_0,\beta_0,\gamma_0}, \\
&= \xi_i^{\alpha_0,\beta_0\gamma_0} + A_i + \widetilde{A_i} + \widetilde{\widetilde{A}}_i,
\end{aligned}
\tag{1}
$$

where

$$
\begin{aligned}
A_i &= \frac{1}{N} \sum_{\gamma \neq \gamma_0} \xi_i^{\alpha_0,\beta_0,\gamma} \sum_j \xi_j^{\alpha_0,\beta_0,\gamma} \xi_j^{\alpha_0,\beta_0,\gamma_0}, \\
&= b_1^2 \sum_{\gamma \neq \gamma_0} \xi_i^{\alpha_0,\beta_0,\gamma},
\end{aligned}
\tag{2}
$$

$$
\begin{aligned}
\widetilde{A}_i &= \frac{1}{N} \sum_{\beta \neq \beta_0} \sum_\gamma \xi_i^{\alpha_0,\beta,\gamma} \sum_j \xi_j^{\alpha_0,\beta,\gamma} \xi_j^{\alpha_0,\beta_0,\gamma_0}, \\
&= b_1^2 b_2^2 \sum_{\beta \neq \beta_0,\gamma} \xi_i^{\alpha_0,\beta,\gamma},
\end{aligned}
\tag{3}
$$

and

$$
\begin{aligned}
\widetilde{\widetilde{A}}_i &= \frac{1}{N} \sum_{\alpha \neq \alpha_0} \sum_{\beta,\gamma} \xi_i^{\alpha,\beta,\gamma} \sum_j \xi_j^{\alpha,\beta,\gamma} \xi_j^{\alpha_0,\beta_0,\gamma_0}, \\
&= 0.
\end{aligned}
\tag{4}
$$

Thus, according to Eq.(7,8) in the main text, we have

$$
C_i = b_1^2 \xi_i^{\alpha_0,\beta_0,\gamma_0} \sum_{\gamma \neq \gamma_0} \xi_i^{\alpha_0,\beta_0,\gamma},
\tag{5}
$$

$$
\widetilde{C}_i = b_1^2 b_2^2 \xi_i^{\alpha_0,\beta_0,\gamma_0} \sum_{\beta \neq \beta_0} \sum_\gamma \xi_i^{\alpha_0,\beta,\gamma},
\tag{6}
$$

$$
\widetilde{\widetilde{C}}_i = 0.
\tag{7}
$$

## 2.2 The distribution of the intra-class noise $C_i$

Define the sets $\mathcal{C}_+ \equiv \{\gamma \mid \xi_i^{\alpha_0,\beta_0,\gamma} = \xi_i^{\alpha_0,\beta_0}, \gamma \neq \gamma_0\}$, and $\mathcal{C}_- \equiv \{\gamma \mid \xi_i^{\alpha_0,\beta_0,\gamma} = -\xi_i^{\alpha_0,\beta_0}, \gamma \neq \gamma_0\}$, and their numbers are denoted as $card(\mathcal{C}_+)$ and $card(\mathcal{C}_-)$, respectively. Note $card(\mathcal{C}_-) = P_\gamma - 1 - card(\mathcal{C}_+)$.

According to Eq.(5) in the main text,

$$P(\xi_i^{\alpha,\beta,\gamma}) = (\frac{1+b_1}{2})\delta(\xi_i^{\alpha,\beta,\gamma} - \xi_i^{\alpha,\beta}) + (\frac{1-b_1}{2})\delta(\xi_i^{\alpha,\beta,\gamma} + \xi_i^{\alpha,\beta}).$$

Since the generalizations of memory patterns are independent to each other, $card(\mathcal{C}_+)$ follows a binomial distribution, $card(\mathcal{C}_+) \sim \mathcal{B}(P_\gamma - 1, P_+)$, with $P_+ = P(\xi_i^{\alpha_0,\beta_0,\gamma} = \xi_i^{\alpha_0,\beta_0}) = (1+b_1)/2$. Here, $\mathcal{B}(n, P)$ denotes a binomial distribution, with $n$ the total number of experiments and $P$ the probability of each experiment yielding a successful result.

In the large $P_\gamma$ limit, a binomial distribution can be approximated as a normal one, which is written as

$$card(\mathcal{C}_+) \sim \mathcal{N}\left((P_\gamma - 1)P_+, (P_\gamma - 1)P_+(1 - P_+)\right). \tag{8}$$

Thus, we have

$$\sum_{\gamma \neq \gamma_0} \xi_i^{\alpha_0,\beta_0,\gamma} = card(\mathcal{C}_+)\xi_i^{\alpha_0,\beta_0} - card(\mathcal{C}_-)\xi_i^{\alpha_0,\beta_0},$$

$$= 2card(\mathcal{C}_+)\xi_i^{\alpha_0,\beta_0} - (P_\gamma - 1)\xi_i^{\alpha_0,\beta_0}, \tag{9}$$

and hence

$$C_i = b_1^2 \left[2card(\mathcal{C}_+) - (P_\gamma - 1)\right] \xi_i^{\alpha_0,\beta_0,\gamma_0} \xi_i^{\alpha_0,\beta_0}. \tag{10}$$

Since the product $\xi_i^{\alpha_0,\beta_0,\gamma_0} \xi_i^{\alpha_0,\beta_0} = 1$ with the probability $(1 + b_1)/2$ and $\xi_i^{\alpha_0,\beta_0,\gamma_0} \xi_i^{\alpha_0,\beta_0} = -1$ with the probability $(1 - b_1)/2$, the distribution of $C_i$ can be written as superposition of two normal distributions, i.e.,

$$P(C_i) = \frac{1+b_1}{2}\mathcal{N}(E_C, V_C) + \frac{1-b_1}{2}\mathcal{N}(-E_C, V_C), \tag{11}$$

where $E_C = b_1^3(P_\gamma - 1)$ and $V_C = b_1^4(P_\gamma - 1)(1 - b_1^2)$.

## 2.3 The distribution of the inter-class noise $\widetilde{C}_i$

Similarly, it can be checked that in the large $P_\gamma$, $P_\beta$ limit, $\widetilde{C}_i$ satisfies superposition of two normal distributions,

$$P(\widetilde{C}_i) = \frac{1+b_1 b_2}{2}\mathcal{N}(E_{\widetilde{C}}, V_{\widetilde{C}}) + \frac{1-b_1 b_2}{2}\mathcal{N}(-E_{\widetilde{C}}, V_{\widetilde{C}}), \tag{12}$$

where $E_{\widetilde{C}} = b_1^3 b_2^3 P_\gamma(P_\beta - 1)$ and $V_{\widetilde{C}} = b_1^4 b_2^4 P_\gamma(P_\beta - 1)(1 - b_1^2)(1 - b_2^2)$.

## 2.4 Retrieval error without feedback

We first calculate retrieval error without the pull-feedback. The probabilities $P(C_i)$ and $P(\widetilde{C}_i)$ can be calculated under four different conditions, which are summarized below:

- When $\xi_i^{\alpha_0,\beta_0,\gamma_0} = \xi_i^{\alpha_0,\beta_0} = \xi_i^{\alpha_0}$, which has the probability $P = (1 + b_1)(1 + b_2)/4$, $P(C_i) = \mathcal{N}(E_C, V_C)$ and $P(\widetilde{C}_i) = \mathcal{N}(E_{\widetilde{C}}, V_{\widetilde{C}})$.

- When $\xi_i^{\alpha_0,\beta_0,\gamma_0} = \xi_i^{\alpha_0,\beta_0} = -\xi_i^{\alpha_0}$, which has the probability $P = (1 + b_1)(1 - b_2)/4$, $P(C_i) = \mathcal{N}(E_C, V_C)$ and $P(\widetilde{C}_i) = \mathcal{N}(-E_{\widetilde{C}}, V_{\widetilde{C}})$.

- When $\xi_i^{\alpha_0,\beta_0,\gamma_0} = -\xi_i^{\alpha_0,\beta_0} = \xi_i^{\alpha_0}$, which has the probability $P = (1 - b_1)(1 + b_2)/4$, $P(C_i) = \mathcal{N}(-E_C, V_C)$ and $P(\widetilde{C}_i) = \mathcal{N}(-E_{\widetilde{C}}, V_{\widetilde{C}})$.

- When $\xi_i^{\alpha_0,\beta_0,\gamma_0} = -\xi_i^{\alpha_0,\beta_0} = -\xi_i^{\alpha_0}$, which has the probability $P = (1 - b_1)(1 - b_2)/4$, $P(C_i) = \mathcal{N}(-E_C, V_C)$ and $P(\widetilde{C}_i) = \mathcal{N}(E_{\widetilde{C}}, V_{\widetilde{C}})$.

The above four conditions can be described by two auxiliary variables, which are $s_1 \equiv \xi_i^{\alpha_0,\beta_0,\gamma_0} \xi_i^{\alpha_0,\beta_0} = \pm 1$ and $s_2 \equiv \xi_i^{\alpha_0,\beta_0,\gamma_0} \xi_i^{\alpha_0} = \pm 1$.

The probability of retrieval error without the pull-feedback is calculated to be,

$$
\begin{aligned}
p(C_i + \widetilde{C}_i < -1) &= \int_{-\infty}^{+\infty} p(C_i = x) p(\widetilde{C}_i < -1 - x) dx \\
&= \sum_{s_1,s_2=\pm 1} \frac{1}{4}(1 + s_1 b_1)(1 + s_1 s_2 b_2) \int_{-\infty}^{+\infty} \frac{dx}{\sqrt{2\pi V_C}} exp(-\frac{x - s_1 E_C}{2 V_C}) \\
&\quad \int_{-\infty}^{\frac{-1-x-s_2 E_{\widetilde{C}}}{\sqrt{2V_{\widetilde{C}}}}} \frac{dy}{\sqrt{2\pi}} exp(-\frac{y^2}{2}), \\
&= \sum_{s_1,s_2=\pm 1} \frac{1}{4}(1 + s_1 b_1)(1 + s_1 s_2 b_2) \int_{-\infty}^{+\infty} \frac{dx}{\sqrt{2\pi}} exp(-\frac{x\sigma_c - s_1 E_C}{2}) \\
&\quad \int_{-\infty}^{\frac{-1-x\sigma_{\widetilde{C}}-s_2 E_{\widetilde{C}}}{\sigma_{\widetilde{C}}}} \frac{dy}{\sqrt{2\pi}} exp(-\frac{y^2}{2}), \\
&\quad (\sigma_C = \sqrt{V_c}; \sigma_{\widetilde{C}} = \sqrt{V_{\widetilde{C}}}), \\
&= \sum_{s_1,s_2=\pm 1} \frac{1}{4}(1 + s_1 b_1)(1 + s_1 s_2 b_2) \int_{-\infty}^{+\infty} \frac{dx}{\sqrt{2\pi}} exp(-\frac{x^2}{2}) \int_{-\infty}^{+\infty} \frac{dy}{\sqrt{2\pi}} exp(-\frac{y^2}{2}), \\
&\quad (\sigma_C x + \sigma_{\widetilde{C}} y < -1 - s_1 E_C - s_2 E_{\widetilde{C}}) \\
&= \sum_{s_1,s_2=\pm 1} \frac{1}{8}(1 + s_1 b_1)(1 + s_1 s_2 b_2) \left[ 1 + erf(\frac{-1 - s_1 E_C - s_2 E_{\widetilde{C}}}{\sqrt{2(\sigma_C^2 + \sigma_{\widetilde{C}}^2)}}) \right]. \quad (13)
\end{aligned}
$$

# 3 The effect of pull-feedback

## 3.1 The distribution of the noise $C_i^*$ after applying the pull-feedback

According to the definition, $C_i^* = C_i - b_1 \xi_i^{\alpha_0,\beta_0,\gamma_0} \xi_i^{\alpha_0,\beta_0}$. It can be checked that for $\xi_i^{\alpha_0,\beta_0,\gamma_0} \xi_i^{\alpha_0,\beta_0} = 1$, the mean $E(C_i^*) = E_C - b_1$, and for $\xi_i^{\alpha_0,\beta_0,\gamma_0} \xi_i^{\alpha_0,\beta_0} = -1$, the mean $E(C_i^*) = -E_C + b_1$. Therefore, we have

$$P(C_i^*) \quad = \quad \frac{1+b_1}{2}\mathcal{N}(E_C - b_1, V_C^*) + \frac{1-b_1}{2}\mathcal{N}(-E_C + b_1, V_C^*), \quad (14)$$

where $E_C = b_1^3(P_\gamma - 1)$ and $V_{C^*} = V_C = b_1^4(P_\gamma - 1)(1 - b_1^2)$.

## 3.2 The retrieval error with the pull-feedback

We now calculate retrieval error with the pull-feedback. Given $W_{ij}^{1,2} = -b_1 \delta_{ij}$ and that layer 2 is at the parent pattern $\mathbf{x}^2(0) = \xi^{\alpha_0,\beta_0}$, the alignment between the neuronal input and the child pattern at layer 1 is written as,

$$\xi^{\alpha_0,\beta_0,\gamma_0} h_i^1(0) \quad = \quad 1 + C_i + \widetilde{C}_i - b_1 \xi^{\alpha_0,\beta_0} \xi^{\alpha_0,\beta_0,\gamma_0}. \quad (15)$$

We obtain

$$m^{\alpha_0,\beta_0,\gamma_0}(1) \quad = \quad \frac{1}{N}\sum_{i=1}^{N} \text{sign}[1 + C_i + \widetilde{C}_i - b_1 \xi_i^{\alpha_0,\beta_0,\gamma_0} \xi_i^{\alpha_0,\beta_0}],$$

$$= \quad \frac{1}{N}\sum_{i=1}^{N} \text{sign}\left[1 + C_i^* + \widetilde{C}_i\right] \quad (16)$$

where $C_i^* = C_i - b_1 \xi_i^{\alpha_0,\beta_0,\gamma_0} \xi_i^{\alpha_0,\beta_0}$.

The probability $P(C_i^*)$ can be calculated under four different conditions, which are,

- When $\xi_i^{\alpha_0,\beta_0,\gamma_0} = \xi_i^{\alpha_0,\beta_0} = \xi_i^{\alpha_0}$, which has the probability $P = \frac{(1+b_1)(1+b_2)}{4}$, $P(C_i^*) = \mathcal{N}[((E_C - b1), V_C]$;

- When $\xi_i^{\alpha_0,\beta_0,\gamma_0} = \xi_i^{\alpha_0,\beta_0} = -\xi_i^{\alpha_0}$, which has the probability $P = \frac{(1+b_1)(1-b_2)}{4}$, $P(C_i^*) = \mathcal{N}[(E_C - b1), V_C]$;

- When $\xi_i^{\alpha_0,\beta_0,\gamma_0} = -\xi_i^{\alpha_0,\beta_0} = \xi_i^{\alpha_0}$, which has the probability $P = \frac{(1-b_1)(1+b_2)}{4}$, $P(C_i^*) = \mathcal{N}[-(E_C - b1), V_C]$;

- When $\xi_i^{\alpha_0,\beta_0,\gamma_0} = -\xi_i^{\alpha_0,\beta_0} = -\xi_i^{\alpha_0}$, which has the probability $P = \frac{(1-b_1)(1-b_2)}{4}$, $P(C_i^*) = \mathcal{N}[-(E_C - b1), V_C]$.

Analogous to Eq.(13), the probability of retrieval error with the pull-feedback is calculated to be,

$$p(C_i^* + \tilde{C}_i < -1) = \sum_{s_1,s_2=\pm1} \frac{1}{8}(1 + s_1 b_1)(1 + s_1 s_2 b_2) \times$$

$$\left[1 + erf(\frac{-1 - s_1(E_C - b_1) - s_2 E_{\tilde{C}}}{\sqrt{2(\sigma_C^2 + \sigma_{\tilde{C}}^2)}})\right]. \quad (17)$$

# 4 The effects of push-pull feedback

## 4.1 Dynamics of the continuous Hopfield model

The dynamics of the continuous Hopfield model are given by

$$\tau \frac{dh_i^1}{dt} = -h_i^1 + \sum_j W_{ij}^1 x_j^1 + \sum_k W_{ik}^{1,2}(t)x_k^2 + I_i^{ext}, \quad (18)$$

$$\tau \frac{dh_i^2}{dt} = -h_i^2 + \sum_j W_{ij}^2 x_j^2 + \sum_k W_{ik}^{2,1}(t)x_k^1 + I_i^{ext}, \quad (19)$$

$$x_i^n = \frac{2}{\pi} \arctan(8\pi h_i^n), \quad \text{for} \quad n = 1, 2, \quad (20)$$

where $h_i^n$ and $x_i^n$ denote the synaptic input and the firing rate of neuron $i$ in layer $n$. $\tau$ is the time constant, $I_i^{ext}$ the external input to layer 1, and $arctan(x)$ is the inverse of the tangent function.

In the continuous model, each element of a memory pattern (e.g., $\xi_i^{\alpha,\beta,\gamma}$) takes a value of 0 or 1. The recurrent connections between neurons in the same layer are constructed by the Hebbian covariance learning rule, which are given by

$$\tilde{W}_{ij}^1 = \frac{1}{N} \sum_{\alpha,\beta,\gamma} \left(\xi_i^{\alpha,\beta,\gamma} - \langle\xi\rangle\right)\left(\xi_j^{\alpha,\beta,\gamma} - \langle\xi\rangle\right), \quad (21)$$

$$\tilde{W}_{ij}^2 = \frac{1}{N} \sum_{\alpha,\beta} \left(\xi_i^{\alpha,\beta} - \langle\xi\rangle\right)\left(\xi_j^{\alpha,\beta} - \langle\xi\rangle\right), \quad (22)$$

$$W_{ij}^n = a_r^n \frac{\tilde{W}_{ij}^n}{|\mathbf{W}^n|}, \quad \text{for} \quad n = 1, 2, \quad (23)$$

where $\langle\xi\rangle$ is the mean activity of all neurons averaged over all memory patterns in the same layer, $|\tilde{\mathbf{W}}^n| = \sqrt{\sum_{ij}(\tilde{W}_{ij}^n)^2/N^2}$, and $a_r^n$ are positive constants.

We consider the feedback connections from layers 2 to 1, which vary over time. At the early phase, the feedback is positive, which is written as

$$W_{ik}^{1,2} = \frac{a_+}{NP_\gamma} \sum_{\alpha,\beta,\gamma} (\xi_i^{\alpha,\beta,\gamma} - \langle\xi\rangle)(\xi_j^{\alpha,\beta,\gamma} - \langle\xi\rangle), \quad (24)$$

where $a_+$ is a positive number.

At the later phase, the feedback is negative which is written

$$W_{ik}^{1,2} = -a_- b_1 \delta_{ik}, \tag{25}$$

where $\delta_{ik} = 1$ for $i = k$ and $\delta_{ik} = 0$ otherwise, and $a_-$ is a positive number.

Let us consider the memory pattern to be retrieved at layer 1 is $\xi^{\alpha_0, \beta_0, \gamma_0}$. The external input to the first layer, which conveys the memory pattern information, is set to be

$$I_i^{ext} = a_{ext} \tilde{\xi}_i^{\alpha_0, \beta_0, \gamma_0} + \sigma \eta_i, \tag{26}$$

where $\eta_i$ is a random number uniformly distributed in the range of $(-1, 1)$, representing the memory-independent noise, and $\sigma$ controls the noise strength. $a_{ext}$ is a positive number. The pattern $\tilde{\xi}^{\alpha_0, \beta_0, \gamma_0}$ is the noise-corrupted signal, which is constructed as follows: starting from the clean memory pattern $\xi^{\alpha_0, \beta_0, \gamma_0}$, we first randomly select $\lambda_1 N$ number of neurons, with $0 < \lambda_1 < 1$, and change their values to match the pattern $\xi^{\alpha_0, \beta_0, \gamma'}$, with $\gamma' \neq \gamma_0$, which represents the intra-class noise. We then randomly select $\lambda_2 N$ number of neurons, with $0 < \lambda_2 < 1$, and change their values to match the pattern $\xi^{\alpha_0, \beta', \gamma_0}$, with $\beta' \neq \beta_0$, which represents the inter-class noise.

## 4.2 Comparing network performances with varying correlation amplitudes

Fig.S1 shows how the retrieval improvement due to the push-pull feedback varies with the correlation $b_1$ between child patterns. We see that the push-pull feedback works very well for a wide range of correlation amplitudes. We also note that it has little effect when the correlation $b_1$ is very small, and may worsen retrieval when $b_1$ is too large. This is understandable, since for a very large $b_1$, the strong negative feedback may shut-down the neural activity at layer 1 (see Eq.(10) in the main text).

# 5 Pre-processing real images with a deep neural network

The dataset, chosen from ImageNet, consists of two types (cat and dog), and each type is made up of 9 sub-types. They are: 1) sub-types of cat: Abyssinian Cat, Angora Cat, Burmese Cat, Egyptian Cat, Manx Cat, Persian Cat, Siamese Cat, Tiger Cat, Tortoiseshell; 2) sub-types of dog: Saint Bernard, Beagle Dog, Border Collie, Cirn terrier, Cardigan Dog, Eskimo Dog, Mastiff Dog, Pug Dog. We chose 100 images per sub-type. We re-scaled each image, with its width and height to be $(256, 256)$, and pixel values in the range of $[0, 255]$. After that, we presented each image as an input to the VGG16(the VGG16 itself was pre-trained by ImageNet) and took the neural activity before the reading-out lay as the

Figure S1: The retrieval improvement due to the push-pull feedback vs. the correlation strength $b_1$ between child patterns. Both the retrieval accuracies with and without feedback are taken at the moment of $5\tau$. The results are obtained by averaging over 20 trials. The child pattern is $\xi^{1,1,1}$. Other parameters are the same as in Fig.3 in the main text.

neural representation of the image, referred to as $\hat{x}$ hereafter. Subsequently, we normalized neural activities, i.e.,

$$\widetilde{x}_i = log(1 + \hat{x}_i), \tag{27}$$

$$x_i = ReLu(\frac{\widetilde{x}_i - <\widetilde{x}_i>}{\sigma_{\widetilde{x}_i}}), \tag{28}$$

where $ReLU(\cdot) = max(0, \cdot)$ is the rectified linear function, and $<\widetilde{x}>$ and $\sigma_{\widetilde{x}}$ are the mean and standard deviation of neural activities over all neurons for each image.

Finally, we generated memory patterns for the hierarchical network. For a child pattern, it is given by

$$\xi_i^{\beta_0,\gamma_0} = \text{sign}(<x_i>_{\beta_0,\gamma_0}), \tag{29}$$

where the average is over all the images belonging to the same sub-type of animal.

The corresponding parent patterns are constructed to be

$$\xi_i^{\beta_0} = <\xi_i^{\beta_0,\gamma}>_\gamma, \tag{30}$$

where the average is over all child patterns belonging to the same parent.