[Reviews · NeurIPS 2019]

Reviewer 1



Update: I apologize for my confusion about the dynamics. I feel more positively now about this work, and have increased my score. 1) For this work to have a lot of value to neuroscientists, I'd want to understand the realism of the dynamics much better. There are two issues here to be addressed: a) how realistic is it for the dynamics of the feedforward pass + recurrence within layers to run to convergence *before* sending down the top-down feedback? What happens if these are concurrent processes, such that units get both bottom-up and top-down inputs at the same time? b) The discussion about inhibitory feedback being delayed relative to excitatory is correct, but suggests a tight constraint: the biology data tell us how much time elapses between these two feedback types. Given the time-scale of the recurrent dynamics in cortex, the authors could then ask (in their model) whether this delay is "enough" for their push-pull mechanism to work. If yes, that would strengthen the result a fair bit. 2) There are other students of hierarchical RNNs, and it is worth discussing the details of those works (there are many) to highlight the novelty of this paper. So, for example, the 2nd sentence of introduction is correct (most DNNs are just feedforward) but a bit misleading (because there are several recent papers about hierarchical RNNs). 3) If the unit activities are sign(input) (e.g., Eq. 1), couldn't the pattern overlap (eq. 6) be -1, in the extreme case where two patterns are opposite? In that case, I disagree with the inequality given after Eq. 6, of 1>m>0. This could be easy to fix (or, in case I am wrong, please just tell me why). 4) It's neat that the dynamics with feedback resemble the monkey data on success trials, and the dynamics without feedback resemble those on unsuccessful ones (no contour recognition). 5) I don't understand how the system is implemented for the natural images. It seems to me like, to define the feedforward connectivity, you need to know the "parent" and "grandparent" patterns corresponding to the "child" patterns. That's straightforward (by construction) in the first part of the paper, where you define distributions for these patterns. But for the natural images, I don't understand how you know the parent and grandparent patterns. E.g. what is the "parent" pattern for the dog images? I get that it has something to do with dogs, but I don't see how you determined the specific numerical values.

Reviewer 2



The paper aims to investigate the role of feedback connections in hierarchal information retrieval. It specifically looks into the potential role of push-pull feedback mechanism in aiding the retrieval process. The authors do this by first formulating a toy model that illustrates how the push and pull feedback helps in reducing inter-class and intra-class noise respectively. Then they show simulation results using real images that further corroborates their intuition. This is an interesting idea and a reasonable attempt at mathematizing a widely observed phenomenon in biology. I think the work would benefit from a more thorough set of experiments and explanations. Strengths: – The paper provides a novel perspective on how the push-pull feedback mechanism could possibly aid hierarchical information retrieval. This could be a potential first step towards building intuitions for understanding the role of feedback in predictive coding. – In addition to the paper’s contribution in the direction of understanding push-pull feedback from a neuroscience point of view, this work also shows a possible advantage of incorporating dynamic feedback connections in deep neural networks for improving information retrieval performance. – The paper addresses the conflicting consequences of correlation in neural encoding: higher correlation is essential for embedding categorical relationship between objects, but higher correlation between neural encodings could degrade information retrieval. Substantial weakness: – The authors distinctly claim that push feedback helps in reducing inter-class noise and pull-feedback helps in reducing intra-class noise. But Fig.S2 suggests that only-pull feedback plays a much greater role than only-push feedback in reducing the inter-class noise. It also seems like the push feedback plays a very minor role in improving the performance when both push-pull is used, as compared to just pull. Thus their argument about both push AND pull feedback seems like it fails to explain the actual empirical findings. In other words, pull-only is almost as good as Push-pull, and within error bars. So the claim that push is helpful is a bit dubious. – When push only is applied and then removed, I expect that the coarse-scale class is improved. They should demonstrate this, even if the fine scale classes have additional errors. – It is unclear how they would handle multiple dimensions of parent categories (white animals vs black animals, and not just cats vs dogs) – Confusing notation. – The justification for the push feedback is clear, but I don't understand the mechanism of the pull improvement. This could use substantially more explanation. Given that the pull feedback is linear, I don't see how it could help the actual inputs be distinguished from the average of a parent class. That seems like it would need to impose an energy maximum at the average children of a given parent, and this requires an energy decreases on either side of the maximum, inconsistent with a linear feedback term. Minor weaknesses: – As mentioned in lines 146-149, the mathematical analysis to show the role of push and pull feedback was restricted to the specific case where a parent pattern is perfectly retrieved already. It would be really insightful if its generalization to other cases were also analytically shown, in addition to the empirical analysis provided in the paper. This may be substantially harder, so it is not necessary. – It would be really informative if the authors could shed some light on how they arrived at the forms of push and pull feedback proposed in equations 9 and 10. Or one step further, they could investigate a richer family of forms that push and pull feedback could possibly take. – Figures have been scaled to non-unity aspect ratio, so fonts and labels are messy. – Their push-pull explanation derives from inter-class versus intra-class variability. This structure is created by their generative model. But other stimulus ensembles have more structure than just inheritance. Would this create more mechanisms than just push and pull? ——After author feedback This is somewhat confusing, because now in this simulation with fewer patterns, pull feedback doesn't matter much, and yet the difference in pattern number is only 2.5x. Typical natural scene statistics will include huge numbers of possible images, categories, etc, which according to the authors should make push feedback irrelevant in natural conditions. I thank the authors for their explanation of the pull mechanism. I have increased my score slightly, from 5 to 6.

Reviewer 3



Although interesting, the paper only shows results with simple datasets or examples. The practical implication is not apparent. The math expressions are not hard to follow without going into the supplementary information.

[Author Response · NeurIPS 2019]

## Replies to Reviewers

We acknowledge the valuable and encouraging comments of the reviewers. Limited by space, we have to only focus on clarifying the main concerns of the reviewers.

**About the realism of the model dynamics**

We actually already considered the concurrent dynamics in our model, see Eq. 12, where a neuron receives the feedforward, recurrent, and feedback inputs concurrently. This model generates the results in Fig. 3&4. Our model does not require the convergence of the recurrent dynamics of a layer before applying the top-down feedback.

The experimental data indicates that the time elapse between two feedbacks is about $10 \sim 20$ ms, which guides us to set the separation between two feedbacks to be $1 \sim 1.5\tau$, with $\tau = 10 \sim 20$ ms the membrane time constant, see the lower panels in Fig. 3&4. We also investigated how the model performance varies with the elapse time, see Fig. S3, which shows that the biological elapse time is adequate for the push-pull feedback to function.

We hope we have addressed reviewer 1's major concern about "the realism of dynamics".

**About the role of push feedback and the working mechanism of pull feedback**

First of all, we would like to apologize for the confusing notations in Fig. S2, where the intra- ($\lambda_1$) and inter- ($\lambda_2$) class noises refer to the external input noises (see the definitions in the last paragraph of Sec. 4.1 in SI at page 8), which may be confused with the interference noises due to pattern correlations used in the main text (coming from $b_1$ and $b_2$).

Indeed, as pointed out by the reviewer 2, if only push is applied, the retrieval of the coarse-scale class is improved, which is demonstrated in Fig. 2A. Notably, the contribution of push feedback varies with the parameters. Fig. S2 are the cases where the contribution of push feedback is minor. By choosing different parameters, we can obtain that the contribution of push feedback is large. An example is shown in the right-hand side figure, in which the number of patterns $P_\gamma P_\beta = 40$ is much smaller than $P_\gamma P_\beta = 100$ used in Fig. S2. In general, we find that when the number of patterns is fewer (or the duration of the external input is short), the push feedback tends to have a larger contribution. Thus, the role of feedback is very important. Consider information retrieval in a deep hierarchical network, where the numbers of high-level patterns in the top layers are fewer, then the push feedback is crucial to achieve good retrieval performances of high-level patterns, which subsequently enhance the retrieval of low-level patterns layer by layer.

Figure 1: Comparing network performances with the push-pull, only push, and only pull feedbacks, respectively, with $P_\beta = 2$, $P_\gamma = 20$, and other parameters the same as in the figures in our paper.

As the push feedback is to enhance the retrieval of sibling patterns from the same parent, it is natural to set its form as the product between the child and parent patterns according to the Hebbian rule. For the pull feedback, we arrive at the current form as it de-correlates sibling patterns (see lines 169-170 in the main text and Sec. 3.3 in SI). We further theoretically prove that this form of pull feedback guarantees to improve the retrieval accuracy (Sec. 3.4 in SI). We may understand the working mechanism of pull feedback intuitively in the following way. By subtracting the common part, it highlights the subtle differences between sibling patterns. For example, the fractional difference between two numbers 101 and 99 is small; but after subtracting the mean 100, we get 1 and -1, whose difference appears to be significant, and the nonlinear threshold-like sigmoid function in the neural dynamics (Eq.13) helps to amplify this difference.

We hope that we have addressed reviewer 2's concerns about the role and mechanism of push and pull feedbacks.

**About practical applications**

Actually, we are now working on applying the push-pull feedback to practical applications and have obtained encouraging preliminary results. Here, we introduce the basic idea. We trained a hierarchical prototypical network (a generalization of the prototypical network) using real images, and obtain hierarchical representations (so-called prototypes) of objects across layers (by this, the child, parent, and other higher-level patterns are learned from data). Since the categorization of objects in the prototypical network is based on their distance in the representational space, we can construct the recurrent connections based on the Hebbian rule in each layer with little distortion to the training results. The neural representations in different layers hold the hierarchical correlation structure as considered in this study. We can therefore add the push-pull feedback in the network dynamics to realize robust and flexible, rough-to-fine information retrieval.

We hope that we have addressed reviewers' concern about the potential practical applications of this study.

[Meta-Review · NeurIPS 2019]

The manuscript studies the role of feedback connections in pattern retrieval networks. This is done in a hierarchical Hopfield-type network. Then, different types of top-down feedback are investigated. The study addresses an important question in computational neuroscience. All reviewers found the results interesting. Besides its relevance to modelling of biology, it is also of potential interest for technical applications in hierarchical information retrieval. Two weaknesses of the manuscript is that it is only applied to relatively simple datasets, and the roles of the feedback components (push-pull) is not entirely clear. Nevertheless, due to the interesting approach, we decided that the manuscript is of potential interest for the NeurIPS audience, in particular for the biology-oriented community.